# The Promoter Region of the Proto-Oncogene MST1R Contains the Main Features of G-Quadruplexes Formation

**DOI:** 10.3390/ijms232112905

**Published:** 2022-10-26

**Authors:** Coralie Robert, Julien Marquevielle, Gilmar F. Salgado

**Affiliations:** 1ARNA Laboratory, Inserm U1212, CNRS UMR 5320, University of Bordeaux, 146 Rue Léo Saignat, F-33000 Bordeaux, France; 2Institut Européen de Chimie et Biologie, UAR 3033 US001, CNRS, University of Bordeaux, 2 Rue Robert Escarpit, F-33600 Pessac, France

**Keywords:** MST1R, promoter, G-quadruplex

## Abstract

MST1R (RON) is a receptor of the MET tyrosine kinase receptor family involved in several cancers such as pancreas, breast, ovary, colon, and stomach. Some studies have shown that overexpression of MST1R increases the migratory and invasive properties of cancer cells. The promoter region of the oncogene MST1R is enriched in guanine residues that can potentially form G-quadruplexes (G4s), as it was observed in other oncogenic promoters such as KRAS and c-MYC. There is abundant literature that links the presence of G4s in promoter regions of oncogenes to diverse gene regulation processes that are not well understood. In this work, we have studied the reverse and forward sequence of MST1R promoter region using the G4Hunter software and performed biophysical studies to characterize the best scored sequences.

## 1. Introduction

Tyrosine kinase receptors (RTKs) are known to be involved in cancer cell proliferation, invasion and metastasis formation [1,2]. The macrophage-stimulating RTK receptor 1 or MST1R (Figure 1A), also known as RON (Récepteur d’Origine Nantais), gathered great interest [3,4,5] concerning the fight against certain types of cancers such as pancreas, breast, ovary, colon, and stomach. The gene encodes for a 180 kDa receptor belonging to the epithelial-mesenchymal transition (EMT) factor family involved in the formation of metastases. Several oncogenic signaling pathways are located downstream of MST1R, including RAS/mitogen-activated protein kinase (MAPK) and phosphatidyl inositol-3 kinase (PI-3K)/AKT. In addition, it is also implicated in the activation process of the focal kinase adhesion pathway (FAK) corresponding to signaling cascades involved in adhesion, cell motility, proliferation and apoptosis [4,6,7]. As common in different cell surface receptor families, RON dimerizes in the presence of its ligand, the macrophage-stimulating protein (MSP). In fact, inflammatory environments such as tumors are enriched in macrophages thus generating high levels of reactive oxygen and nitrogen species. Degraded macrophages (pro-MSP) can bind to MST1R and thus activate the signal transduction pathways involved in the propagation of cancer cells. Studies have shown that overexpression of MST1R in colon, ovarian and breast cell lines increases the migratory and invasive properties of cancer cells [8,9,10] by inducing intraepithelial neoplasia in both human and mouse primary metastatic cell lines [11] thus asserting that MST1R is indeed involved in pancreatic cancer [12]. In addition, overexpression of MST1R is also common in patients with stomach cancer [5].

Despite the numerous molecular agents developed in the last decades such as trastuzumab, lapatinib or bevacizumab that were tested and approved to target RTKs [13,14], lack of selectivity and the appearance of numerous side effects left the medical community without any robust and safe drugs to block RTKs. Nevertheless, targeting MST1R alone or in a concerted fashion could be an effective therapeutic strategy for the treatment of certain cancerous profiles, including pancreatic and stomach cancers. Therefore, it is important to look for new directions and find ways of targeting this receptor family. The receptor is encoded by the oncogene MST1R located on chromosome 3 in humans (49887002–49903873) on the antisense strand. This gene can produce different transcripts due to alternative splicing during transcription that will, or not, code for different isoforms of the MST1R receptor. The promoter region of the oncogene MST1R consists of a classical promoter sequence whose role is fundamental for transcription. This region is rich in guanine bases that can potentially form G-quadruplexes (G4s) (Figure 1B). G-quadruplexes are unusual nucleic acid structures in a “world” dominated by canonical duplex DNA. These are three-dimensional (3D) structures based on four-stranded secondary domains that can be adopted by nucleic acids rich in guanine residues. They result from the stacking of at least two guanine tetrads connected by Hoogsteen-type hydrogen bonds between four coplanar guanines and stabilized by monovalent cations such as potassium [15,16]. Thus, the aim of this research work is to probe by biophysical methods the G-rich sequences found on the oncogene MST1R promoter region (Figure 1A) and try to understand if they would be promising targets for developing small ligands against them as it has been done to other G4s from different promoter regions such as KRAS and c-MYC [17,18,19,20]. The presence of G4s in a variety of promoter regions of oncogenes is known to be directly connected with regulation mechanisms [15]. The production of messenger RNAs can be severely reduced and the expression of mutated proteins at the origin of certain cancers mitigated. With that objective in mind, we have studied five reverse and five forward sequences found within the promoter region by analyzing the G4 rate that can potentially be formed in this sequence using the G4Hunter software [21]. In addition, we analyzed these sequences by Nuclear Magnetic Resonance (NMR) and Circular Dichroism (CD) spectroscopies, UV-melting and native gel electrophoresis.

## 2. Results and Discussion

### 2.1. MST1R Promoter Contains G4 Forming Sequences

Using the G4Hunter software, we selected 10 sequences (Appendix A) in the promoter region of the MST1R oncogene with high scores with potential to form G4s (Table 1). This ability to form G4s is characterized by a score assigned automatically by the software. According to Bedrat et al., 2016, sequences which have a score between 1 and 1.25 have low probability to fold into stable G4s structures. Above a score of 1.5, the sequences tend to form a stable G4 as described elsewhere [22]. Usually, the stability protocols are performed at 25 °C, and in some cases structures are reported stable at temperatures as high as 37 °C [23].

### 2.2. MST1R Promoter Contains G4 Forming Sequences

In order to assess the global topology, the melting temperature and overall stability, we performed CD experiments on each sequence (Figure 2A). These experiments show that all the analyzed sequences fold into G4s tertiary structures having a characteristic signal of G4s which seems consistent with the scores attributed by G4Hunter software since all the sequences have a score between 1.5 and 2 corresponding to sequences with the propensity to form stable G4s.

However, there are different topologies of G4s depending on each sequence. Some oligonucleotides fold into parallel conformation type characterized by a positive peak at 260 nm and a negative peak at 240 nm, as is the case for sequences R2, R4, R5, F1 and F5. In addition, we also have sequences that form antiparallel G4s, with a characteristic negative peak at 260 nm and a positive peak around 290 nm, as is the case of sequence F2 and a partial antiparallel or mixture for R1 and R3. We can also note hybrid G4s as shown by sequences F3 and F4. The sequences were tested in the presence of KPi buffer with 50 mM KCl. Then we performed UV-melting in order to assess the relative stability parameters for the different G4s formed in these sequences (Table 1, Figure 2B and Appendix A). We can observe that the most stable G4s seem to be the sequences F2 and F4 with melting temperatures of 60.4 °C and 68.9 °C, respectively. In general terms, the results indicate the presence of very stable G4s that accommodate the possibility for their existence at physiological temperatures.

### 2.3. MST1R Promoter Can Form Intramolecular G4s

To obtain information about the molecularity of the sequences in Table 1, we performed native gel electrophoresis experiments (PAGE) (Figure 3). We found out that almost all probed sequences are monomeric species seen by a single compact band characteristic of well-folded species or have smaller less intense bands at higher molecular identities. Overall, these conformations migrate with a band less than 25 bp corresponding to a compact and well-folded G4 specie found, for example, in the R1, R2, R3, F2 and F4 sequences. Sequences R5, F3 and F5 have multiple bands along different marker sizes and a strong band at the well bottom, clearly indicating the presence of multimeric species.

As observed by PAGE, we may have the contribution of a homodimer for F4. In UV-melting experiments, we might not observe a mix of each structure but the disaggregation or the loss of interaction within the homodimer which is not observed in F2 corroborated by its PAGE profile.

In order to further investigate the molecularity and obtain the folding signature characteristic of each oligo sequence, we performed solution-state 1D ^1^H-NMR experiments at 37 °C (Figure 4). The results allow us to confirm the presence of a G4 folding by the characteristic guanine imino peaks in the 1D ^1^H spectra region between ≈10.4 and 12.5 ppm. In addition, by observation of the spectral line width and the number of individual imino peaks we can deduce that some G4s look more polymorphic than others corroborating the native PAGE results. Spectra of R1, R4, F2 and F4 have well-defined and sharp peaks, while the spectra corresponding to R2, R3, R5, F1, F3 and F5 seem to have much broader and undistinguishable set of imino peaks characteristic of a strong polymorphic behavior under the experimental condition. From those, F1, F3 and R5 display the pattern of very complex entanglements with high molecular weight species that do not penetrate the running-part of the polyacrylamide gel. In the case of F3, we found mainly a homodimer specie with a small presence of other minor species. Overall, the lack of intensity can be explained by the CD data which showed that the G4s contained in the F3 and F4 sequences seem to be G4s having a hybrid topology which can therefore explain a very strong polymorphism of these sequences. Previous studies of telomeric DNA hybrid G4s [24] have shown that polymorphism and interconversion between species is expected and occurs at slow exchange rates <µs. In addition, Sugiyama and colleagues [25] have also reported that hybrid structures coexist at slow exchange rate with other species such as triplex of parallel strands which can also refold into an antiparallel chain. The presence of dimers would further complicate the equilibrium between species for samples such as F3.

Furthermore, the poor quality of the NMR signals observed for samples R5, F1, F3 and F5 can be correlated with the enrichment of cytosine bases in these sequences. The increase of C bases can compete for base pairing with guanines, which are important for the tetrads stabilization as shown in Appendix A. Indeed, UV-melting profiles at 260 nm indicate that R5, F1, F3 and F5 can form base pairs. Furthermore, it seems that R1, F2 and F4 are capable of forming base pairs in addition to the G4 structure as observed in NMR profile with GC base pairs (13–14 ppm) for R1 and F4, whereas F2 seems to form AT base pairs (around 14 ppm). In principle, base pairs can be preferable formed internally to each monomer, but they can also occur between two monomeric units, further increasing the polymorphism of some species such as R5. It should be taken into account that inter-monomer interactions are weak and represent different species in the folding landscape. NMR spectra are compatible with duplex DNA motifs and excludes the presence of stable i-motifs at experimental conditions. Nevertheless, we cannot infer the possibility that those sequences enriched in cytosines may be the target of methylases and play a role in gene regulation.

## 3. Materials and Methods

### 3.1. Putative Quadruplex Sequences (PQS) Selection

Analyses were performed on MST1R promoter sequence in the full chromosome sequence downloaded from the National Center for Biotechnology Information (NCBI) database (current reference GRCh38; NCBI ID NC_000003.12). PQS were selected using G4Hunter algorithm which selected all sequences with a score above or close to the selected threshold (1.5) with a window of 20 nucleotides [21]. G4Hunter identified sequences on both positive and negative strands. All the selected sequences have been summarized in Table 1.

### 3.2. DNA Samples Preparation

Oligonucleotides were purchased from Integrated DNA Technologies (IDT) with a desalting purification option. They were washed in water and then in 1× buffer (10 mM K_2_HPO_4_/KH_2_PO_4_; 50 mM KCl; pH 6.6) by successive centrifugation with Amicon 2 kDa MWCO filters (4500 rpm, at 25 °C). The choice of pH was set in order to optimize the protonation of the imino bonds and consequently improved NMR signal in the imino region (≈10.5 to 12.5 ppm). In order to form G-quadruplexes, samples were heated at 95 °C for 5 min then quickly cooled in ice. This annealing process was repeated three times. Without the annealing procedure, we usually observe broader imino peaks in the NMR spectra. Concentration of the final sample was measured by UV-vis spectroscopy. Sample stocks were kept at −20 °C until further use.

### 3.3. Circular Dichroism (CD)

Quartz cuvettes contained 500 µL of 15 μM of oligonucleotide sample in 1× buffer. All experiments were performed with a JASCO J-1500 CD spectrometer. Spectra were measured between 220 and 330 nm with a scan speed of 50 nm/min. Regular CD spectra were recorded at 37 °C.

### 3.4. UV-Melting

All experiments were performed using a SAFAS UVmc2 double-beam spectrophotometer (Monte Carlo, Monaco) equipped with a 10-cell holder regulated by a Peltier controller. The oligonucleotide strand concentration was 5 µM (1× buffer) in 500 µL quartz cuvettes. Melting curves were recorded both ways (heating and cooling) between 90 °C and 10 °C at 0.4 °C min^−1^.

### 3.5. Native Gel Electrophoresis

Native gel experiments were performed at 4 °C in a 15% acrylamide/bisacrylamide (19:1 ratio) gel containing 10 mM KCl and 1× TBE buffer (89 mM Tris-borate and 2 mM EDTA, pH 8.3, Sigma Aldrich, France). A running buffer with 1× TBE and 10 mM KCl was also prepared. Oligonucleotide samples were prepared at 25 μM in a final volume of 20 μL containing 6 µL of sucrose. A low molecular weight DNA ladder from New England Biolabs (range from 25 to 766 bp) was used. A 1 h pre-run was performed under native conditions (4 °C) in order to eliminate impurities. Samples were loaded and migrated for 3 h at 110 V. Oligonucleotides bands were revealed with 0.01% Stains-All dye (C_30_H_27_BrN_2_S_2_, Sigma Aldrich).

### 3.6. Solution NMR 1D ^1^H Spectroscopy

NMR spectra were recorded on a Bruker Advance 400, 700 MHz spectrometer at 310 K. Samples were prepared in 1× buffer (10 mM K_2_HPO_4_/KH_2_PO_4_; 50 mM KCl; pH 6.6) supplemented with 10% D2O added for the lock in a 3 mm tube. The concentration of DNA samples was between 0.3 and 1 mM. In all experiments, a time domain (TD) of 32 K, a spectral width (SW) of 20 ppm, an acquisition time (AQ) of 2 s and a relaxation delay (d1) of 2 s were used. Spectra were processed and analyzed with Bruker TopSpin 4.0.9 software.

## 4. Conclusions

To summarize the present work, we tested whether guanine-rich sequences present at the promoter region of the MST1R oncogene can form G4s structures in vitro. The evidence gathered from CD, PAGE and NMR indicates that six sequences out of 10 tested fold into G4s with a rather stable conformation at 37 °C. However, these sequences seem to adopt different topologies and molecularities, testifying to a possible structural polymorphism. Further studies are necessary to test their presence at the promoter level *in cellulo* and possible effects when perturbed by ligands or by Crispr Cas9 technology. Their existence in vivo can lead to the development of new therapeutic targets by studying with atomic details the interaction with transcription regulatory elements. The development of ligands that target the plausible G4 structures may lead to transcription disruption phenomena and ultimately reduce cell proliferation and metastasis derived from MST1R activation cascade. Currently, other in vitro studies have demonstrated the existence of G4s in certain regions of the promoter sequence of the gene that codes for c-Kit membrane receptor [19] which is also a receptor of the RTK family. There is a consensus that its promoter region has three G4 stretches named kit1, kit* and kit2 [26] and all three sequences are also G-rich. It would therefore be interesting to study each of MST1R G4 sequences in detail and obtain further 2D and 3D structural data for the most relevant structures obtained from a complete set of polymerase stop and cellular viability assays. A detailed study would allow to compare results with other cases where G4 structures are present in oncogene promoter regions such as KRAS [17,18,23] or c-MYC [20], and thus better understand their potential role in transcription regulation. Additional studies where several G4 ligand families could be tested against the abovementioned sequences would also be interesting for comparative studies, having in mind possible therapeutic purposes [27,28].

## Figures and Tables

**Figure 1 ijms-23-12905-f001:**
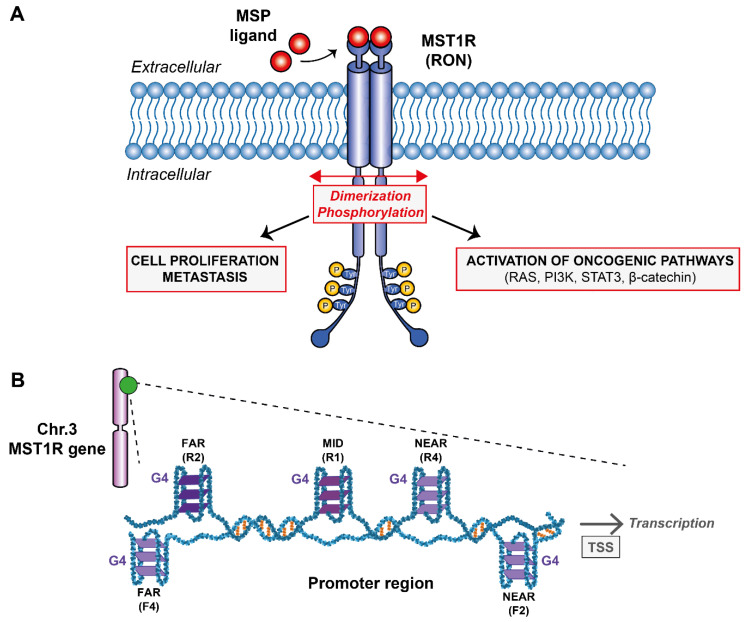
(**A**) Schematic representation of the G-quadruplexes formed within MST1R promoter region and their potential role on transcription. (**B**) Downstream signaling pathways after MST1R activation by dimerization and phosphorylation upon binding of MSP ligand.

**Figure 2 ijms-23-12905-f002:**
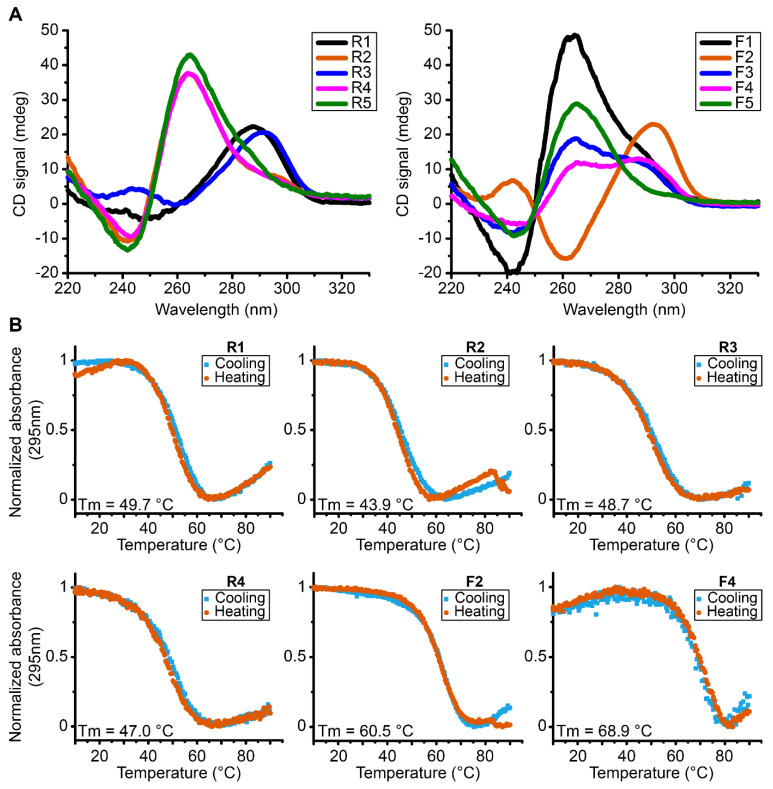
(**A**) CD spectra of reverse and forward MST1R G4 forming sequences at 37 °C exhibiting parallel (R2, R4, R5, F1 and F5), antiparallel (F2), hybrid conformation (F3, F4) and a partial antiparallel or mixture (R1 and R3). (**B**) UV-melting spectra of reverse and forward MST1R G-quadruplexes forming sequences at 295 nm from 10 °C to 90 °C. All experiments were performed in buffer 1× (50 mM KCl; 10 mM KPi; pH 6.6).

**Figure 3 ijms-23-12905-f003:**
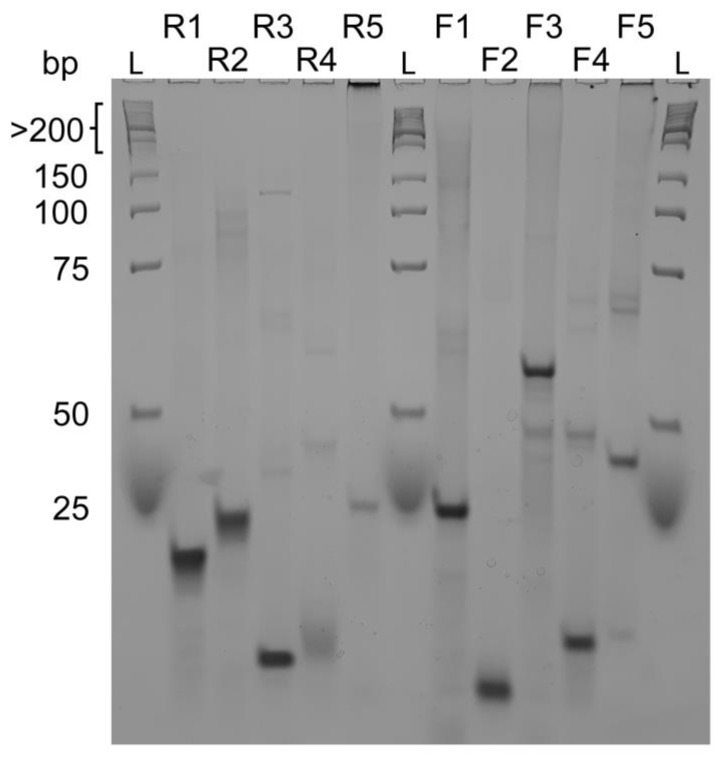
Native gel electrophoresis experiment of reverse sequences R1, R2, R3, R4, R5 and forward sequences F1, F2, F3, F4, F5 compared to a low molecular weight DNA ladder. R3, R4, F2 and F4 seem to form intramolecular G-quadruplexes whereas all other sequences form intermolecular structures. Native gel experiments were performed at 4 °C with 15% acrylamide/bisacrylamide (19:1 ratio) gels.

**Figure 4 ijms-23-12905-f004:**
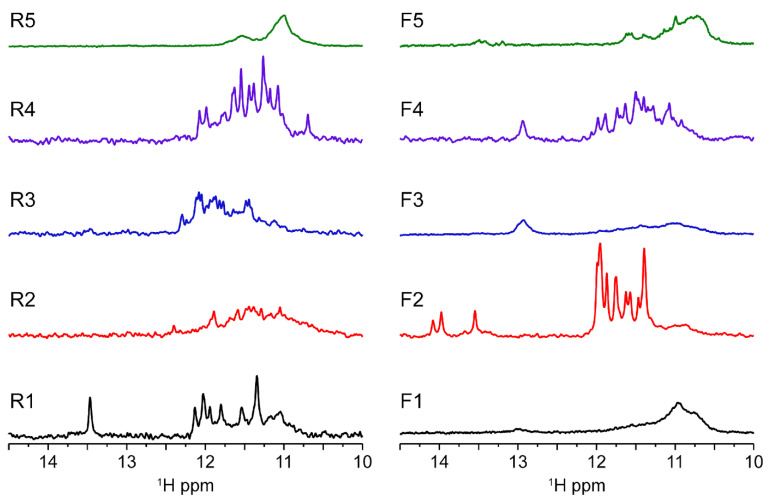
MST1R reverse and forward sequences imino region at 37 °C showing results in agreement with native gel experiment. R1, F2 and F4 exhibit spectra with quite well-resolved peak supporting the formation of three-tetrad G-quadruplexes. Looking to the medium-broad 1D NMR profile, some monomeric species seems to be in equilibrium between different conformers (R3 and F4), whereas the other seems to form higher molecular weight complexes. R1 seems to form a bimolecular quadruplex. All experiments were performed in buffer 1× (50 mM KCl; 10 mM KPi; pH 6.6).

**Table 1 ijms-23-12905-t001:** All MST1R sequences identified by G4Hunter using a window of 20 nucleotides and a threshold of 1.5 with their corresponding score on both reverse and forward strand. Their capacity to form a G4 as well as their melting temperature have been indicated based on UV-melting experiments at 295 nm in 10 mM potassium phosphate buffer pH 6.6 supplemented with 50 mM KCl.

Name	G4Hunter Score	Sequence (5′ → 3′)	ProbabilityG4 Forming Sequence	Tm (°C)
		** *Reverse strand* **		
**R1**	1.66	CGG CTG GGG CGG CAG GTG AGG CGG CTG GGG C	**High**	49.7
**R2**	1.72	AGT CGG TAG TGG GGG GAT GGG ATG GGA CGG C	**High**	43.9
**R3**	1.50	CGG ATG GGC GGA GGG CCT GGG C	**High**	48.7
**R4**	1.60	AGG GCC GGG AAG GGA TTT GGG T	**High**	47.0
**R5**	1.44	AGG AAC CTG GGG CGG GGG TCC GCT ATC TTG GGG	**Low**	47.9
		** *Forward strand* **		
**F1**	1.44	AGG GCG CCG GGC TGG GCG GGC GGA GTC GGG CCG TGG GGG CGG GGC CGC GAG GAA G	**Low**	57.0
**F2**	1.76	TGG GCG TGG GCC TGG CTG GGG GC	**High**	60.5
**F3**	1.44	AGT CGG GCC GTG GGG GCG GGG CCG CGA GGA AGG C	**Low**	57.3
**F4**	1.44	AGG GCG CCG GGC TGG GCG GGC GGA	**High**	68.9
**F5**	1.67	AGA AGG GGG GCA GGA CAC TGG GC	**Low**	<50.0

## Data Availability

Not applicable.

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
