# Peer review of "The Promoter Region of the Proto-Oncogene MST1R Contains the Main Features of G-Quadruplexes Formation"

_ijms, 2022, doi:10.3390/ijms232112905_

Round 1

Reviewer 1 Report

The manuscript by Robert, Marquevielle and Salgado highlights the propensity of DNA fragments around the MST1R proto-oncogene promoter to fold into guanine quadruplexes of various molecularities and topologies. These fragments were identified through a G4Hunter screen. This work adds to the list of G-rich promoter sequences with potential for transcriptional control. Considering the importance of tyrosine kinase receptors in cancer progression, this report is of great interest for scientific communities invested in cancer research.

 Some clarification and expansions would be greatly beneficial before publication.

1) Clarifications

Page 1 (introduction): Can be authors clarify what lies behind the ‘RON’ acronym?

Page 1: The authors may want to rephrase the sentence that deals with ‘the activation process of the focal kinase adhesion pathway…which are signaling cascades’ in order to avoid the singular subject/plural verb grammatical conflict.

Page 1, line before last: Can the authors clarify what is meant by ‘MST1R is well involved in…’? Do they mean ‘MST1R is indeed involved in…’, or ‘MST1R is heavily involved in…’?

Page 3 (end of paragraph 1): For purists, please replace ‘we performed NMR, circular dichroism and native gels’ with ‘we analyzed these sequences by nuclear magnetic resonance (NMR) and circular dichroism (CD) spectroscopies, and native gel electrophoresis’.

Page 3, paragraph 2.1: Is there a need for an apostrophe in ‘predictions’?

Page 3, paragraph 2.1: The authors may want to be less definitive at this point of the manuscript, when they state ‘the sequences tend to form a stable G4, and beyond a score of 2 all sequences form stable G4’ on the sole basis of G4Hunter scores. The formation of G4 can only be confirmed experimentally (so not quite at the G4Hunter score stage).

Page 3, paragraph 2.1, last sentence: ‘reported stable at temperatures as high as 37 C’, rather than ‘in temperatures’?

Page 5, paragraph 2.3, first sentence, as well as Figure 3 captions: ‘native gel electrophoresis experiments’ would be more exact.

Page 6, 3rd line from the top: Can the authors clarify the following statement: ‘F1, F3 and R5 are well behind the NMR time scale’?

Typos/proof-reading requirement:

In 3.2 DNA samples preparation: remove ‘have’ in ‘They have were washed’. Also ‘6,6’ needs to be turned into ‘6.6’. Finally, ‘measured by UV-vis spectroscopy

In 3.4  Native gel electrophoresis, it seems that part of a sentence may be missing, in  ‘in order to eliminate.’, as well as ‘samples were loaded in migrated for 3 hours’.

In conclusion, ‘other studies have raised the…’

In conclusion, ‘the existence of G4s in certain regions of membrane receptors such as c-Kit’. Do the authors mean, ‘the existence of G4s in certain regions of the gene promoter of membrane receptors such as c-Kit’

In Funding: Please revise the sentence acknowledging financial support.

2) Expansions

Page 3, Table 1 caption: Can the authors justify the choice of a pH of 6.6, early (/earlier) in the manuscript? In the latter part, i-motifs are mentioned, so there is likely a connection, but it is not made explicit. Some expansion behind the choice of pH values, and maybe some comparison with (/comment on) similar experiments other, more basic, pH values would be useful to fully capture the authors' work.

Page 4, Figure 2 CD spectra: The text states that R4 (pink) adopts a parallel conformation, while R2 (red) seems to be hybrid. However, the CD curves seem to be super-imposable. Can the authors justify the statement? Would R2 not have a strong parallel component in its CD signature?

Similarly, F1 is listed in the text as hybrid. Although there seems to indeed be a hybrid contribution in the form of a shoulder around 300 nm, doesn’t F1 have a strong parallel contribution (negative peak at 240 nm, and large positive peak at 260 nm)?

Page 4, Figure 2 melting: Can the authors comment on the shape of the melting curve for F4 (slight increase in CD signal in the 1st and last stages of heating)?

Page 5, middle of top paragraph: The authors suggest that ‘a hybrid topology..can therefore explain a very strong polymorphism of these sequences’. Connecting the (thermodynamic) adoption of a topology, and its (kinetic) ability to exchange among various forms, is a very interesting, and intriguing, aspect of G4 self-assembly. Can the authors expand on the relationship between the hybrid topology, and its propensity to be polymorphic? Is this contemplated in other work that may be cited here?

Page 5, end of top paragraph: The very interesting discussion becomes a little bit confusing at the end of this paragraph. First, the discussion aims at confronting G-C pairing as a competing effect to G4 formation, presumably through the formation of duplex segments, which seems a valid point of discussion. The text then turns to the observation of some AT / GC base-pairing based on NMR data. However, such local base-pairing is not necessarily incompatible with G4 formation, as the loop structures of a G4 may, given the appropriate orientation and distances, adopt some base-pairing. The phrasing at this point of the manuscript can therefore be confusing for the reader. Can the authors clarify this portion of the text?

Page 5, end of top paragraph: The sentence ‘In addition, they can induce the formation of homodimers further increasing the sample distribution of less stable conformers’ would also benefit from some clarification. Are homodimers always less stable conformers (or are the authors trying to convey some other aspects here)?

Page 5, Figure 4 caption: seems to suggest that broader spectra always contain multimeric G4s (described as ‘higher weight complexes’ in the text). Is this always true? Can a broad spectrum not be the result of intermediate exchange rates between monomeric G4s? Please clarify.

Page 7, DNA samples preparation: The manuscript describes that the rapid folding process was repeating three times. For the technical interest of the reader, can the authors comment on the requirement to do so? What happens with only one rapid folding cycle with the described sequences?

Once these items addressed, there is no doubt that the described research will be of great interest and inspiration for further developments.

Reviewer 2 Report

In this original article entitled “The promoter region of the proto-oncogene MST1R contains the main features of G-quadruplexes formation”, by Robert et al., investigate the potential of five reverse and five forward sequences within the promoter region to form G4 structures. This work presents useful information, and it could therefore represent a platform for further studies aim at targeting MST1R promoter regions. I do not have any major suggestions. However, several points should be addressed by the Authors before this manuscript is acceptable for publication. 

-       line 16. Change “link” with “links”.

-       Line 25. Figure 1A should be mentioned before Figure 1B. Please change the order of the panels in Figure 1.

-       Line 25. The references 3-5 are not recent. Please remove the word “recently” or updated references.

-       Line 32. The subject of the sentence is “pathway” which is singular. Change to “which is”.

-       Line 37. Remove “RON”. It has been already mentioned before.

-       Line 41. Do the Authors mean mouse models? Please clarify.

-       Line 63. Please find synonym to avoid the repetition of the word “structure”.

-       Line 69. Please rephrase to avoid “worthy studying”.

-       Line 76. Define NMR before using acronym.

-       Line 97. Define CD before using acronym.

-       Figure 2 A and 2 B. Please choose figure colors suitable for color blind people.

-       Line 106. Change 6.66 to 6.6.

-       Line 110. Also sequence R2, R5, F1 and F5 show a pick at 240 and 260. Please clarify.

-       Line 184. Delete “have”

-       Line 191. Add CD after circular dichroism 

-       Line 205. Replace with “25 μM in a final volume of 20 μL..”

-       Line 208. Define what is eliminated.

-       Line 208. Clarify what migrates

-       Line 213 change to “spectrometer”

-       Line 222. Change G-quadruplexes to G4s

-       Line 227. Change “Crisper” to “Crispr”

-       Line 235. G-rich not C-rich
